# The Role of Job Control and Job Demands in Becoming Physically Active during the COVID-19 Pandemic: A Three-Wave Longitudinal Study

**DOI:** 10.3390/ijerph19042168

**Published:** 2022-02-15

**Authors:** Valerie Hervieux, Hans Ivers, Claude Fernet, Caroline Biron

**Affiliations:** 1Department of Management, Université Laval, Québec, QC G1V 0A6, Canada; caroline.biron@fsa.ulaval.ca; 2Center of Expertise for the Management of Occupational Health and Safety, Québec, QC G1V 0A6, Canada; 3Center of Research for Sustainable Health VITAM, Québec, QC G1J 2G1, Canada; 4School of Psychology, Université Laval, Québec, QC G1V 0A6, Canada; hans.ivers@psy.ulaval.ca; 5Department of Human Resources Management, Université du Québec à Trois-Rivières, Québec, QC G8Z 4M3, Canada; claude.fernet@uqtr.ca

**Keywords:** physical activity, job control, job demands, psychosocial working conditions, COVID-19 pandemic

## Abstract

Organizational studies suggest that certain psychosocial working conditions are liable to foster positive health outcomes, such as engaging in leisure-time physical activities. However, the psychosocial factors contributing to this improvement remain unexplored, particularly in the workplace and in the context of the decline observed in the physical activity level of the population worldwide. The objective of the study was to examine whether exposure to different combinations of psychosocial working conditions during the COVID-19 pandemic predicts the probability of becoming physically active among Quebec workers. Job demands, job control, and physical activity were assessed three times during the first year of the pandemic via an online questionnaire among physically inactive workers (*n* = 440). Logistic regression analyses were conducted to examine the associations between various combinations of psychosocial risks and physical activity. A total of 117 participants became physically active during the study. After controlling for covariates, active jobs increased the odds of becoming physically active, compared to high-strain jobs (OR = 2.57 (95% CI 1.13 to 5.87)). Having a highly demanding job may not negatively impact physical activity if workers have enough job control to achieve the required tasks.

## 1. Introduction

To foster behaviors that can benefit employee health, positive and healthy working conditions are particularly important, especially in the context of the pandemic. There is considerable evidence suggesting that psychosocial working conditions, such as low job control and high job demands, are important predictors of job strain [1,2]. However, when job demands are balanced against individual and organizational resources [3], this can not only protect workers from strain but also foster positive health outcomes [4], such as engaging in leisure-time physical activities [5,6,7,8]. Several studies have also provided consistent support for the hypothesis that unfavorable work characteristics have a negative effect on physical activity [9,10,11,12], which is generally defined as any bodily movement produced by skeletal muscles that results in energy expenditure [13]. Most of the studies that have explored this relationship focused on the negative side, such as the effect of job strain and adverse health behaviors (i.e., physical inactivity) (see Fransson et al. 2012 for a meta-analysis). However, the psychosocial work factors contributing positively to health (i.e., becoming physically active) remain unexplored.

This is an important concern for many reasons. Firstly, during the COVID-19 pandemic, a decline in physical activity levels has been observed in several studies in diverse populations [14,15,16,17]. However, workplaces can help to promote physical activity by either promoting it through interventions [18] or by providing healthy psychosocial conditions for employees [5,6,7,8]. Secondly, maintaining regular physical activity is even more important during a pandemic context in order to reduce the risks related to chronic diseases. Indeed, physical activity reduces the risk of cardiovascular disease (including heart disease and stroke), hypertension, diabetes, and obesity by improving physical function [19]. There is ample evidence showing that people with such chronic conditions are more vulnerable and at risk of experiencing severe symptoms of COVID-19 [20,21]. Being active helps to prevent those vulnerabilities that put people at risk in the context of this pandemic, reduces costs for health systems and prevents workers from being absent from work for prolonged periods [22,23]. Lastly, considering the acute labor shortages in several and sensitive work sectors during the pandemic [24], it is crucial to understand how psychosocial conditions might foster healthy behaviors to create healthier workplaces [24]. Despite the negative effects of the pandemic in many aspects of life, it has been suggested that the crisis also presents opportunities for growth, change, and the adoption of new behaviors [25].

According to the job demand-control (JDC) model [26], job control fosters health improvements by offering workers more opportunities for action, especially in the presence of high job demands. Although several studies have documented the relevance of the JDC model in relation to health outcomes [27,28], very few have focused on its longitudinal association with the probability of becoming physically active. Considering that physical activity levels decreased in several subgroups during the pandemic [14,15,16,17], it is important to understand how workplaces can contribute positively to the development of healthy lifestyle habits. This study aims to examine whether exposure to different combinations of job demands and job control predict the probability of becoming physically active during the first 8 months of the COVID-19 outbreak. This is an important theoretical issue, especially in the context of the COVID-19 pandemic. From a practical point of view, workplaces can be mobilized both to implement work organization methods that promote physical activity and to encourage employees and managers to be physically active.

### Job Demand-Control (JDC) Model

The JDC model [26] focuses on two job characteristics: demands and control. Job demands refer to psychological stressors such as work overload, time pressure, and conflicting demands [29]. Job control (also called decision latitude) characterizes the ability to make decisions (decision authority) and the opportunity to use specific job skills in the work process (skill discretion) [29].

The combination of job demands and control suggests four quadrants of jobs: high-strain jobs (high demands and low control), active jobs (high demands and high control), passive jobs (low demands and low control), and low-strain jobs (low demands and high control). Thus, JDC is based on the premise that job control not only buffers the development of psychological and physical strain when job demands are high but also foster positive work-related (e.g., a sense of professional accomplishment) [30] and health outcomes (e.g., a physically active lifestyle) [9]. Workers who have a high level of control over their work may have more time to plan leisure-time physical activities or more opportunities to adjust their work time to engage in physical activity [31,32]. 

Some longitudinal and experimental studies [7,33,34] have observed a negative effect of high-strain jobs on the level of physical activity, whereas other studies did not support this association [8,35] or only found a weak association [9]. For example, the results of a four-wave longitudinal study by Oshio et al. [7] among a large Japanese sample (*n* = 9871) provided evidence that the risk of being physically inactive was 22% higher for participants in high-strain jobs and 17% higher for participants in active jobs than for participants in low-strain jobs over time. In contrast, the results of a longitudinal study by Kouvonen, Vahtera [9] reported only a weak association between chronic exposure to low job control and the risk of being inactive. Participants who reported an increase in strain were slightly more likely to be inactive at follow-up than those who reported a decrease in strain. Other studies [11,12] have shown that passive jobs are also associated with a physically inactive lifestyle. This may be explained by a motivational pathway whereby the non-stimulating nature of passive jobs induces a more passive lifestyle through lower self-efficacy [36]. On the contrary, and although appealing, the active job hypothesis (i.e., high control and high demands) in relation to physical activity has only been partially supported by previous studies.

However, despite the recent research interest in this area [37], possible explanations for these inconsistent findings include the lack of studies with a longitudinal design [35] and the difficulty of controlling for individual worker characteristics [38]. Häusser and Mojzisch [37] argue that “[…] predicting well-being from job characteristics and predicting well-being from physical activity have not been connected systematically, despite their mutual aim to identify and describe antecedents of reduced well-being and impaired health” (p. 2). To address the lack of structure and theory to explain the relationship between job characteristics (or psychosocial working conditions) and physical activity, Häusser and Mojzisch [37] proposed a model inspired by the JDC model. In their conceptual article, they propose the “physical-activity-mediated demand-control (pamDC) model” as a new theoretical framework. The two key ideas of the pamDC model are that (1) demands and control act directly and indirectly on physical activity through self-regulation and feelings of self-determination, and that (2) physical activity mediates, in return, the effects of demands and control. 

To our knowledge, only one recent experimental study has tested the causal effects of job demands and job control on physical activity [33]. Abdel Hadi et al. [33] led two experiments in which they simulated a call center with customer interactions and calculation tasks that were followed by a physical activity task (i.e., cycling on a stationary bicycle). In both experiments (*n* = 251), job demands were manipulated (high vs. low) between subjects in terms of customer unfriendliness and task difficulty. In one of the experiments, researchers manipulated job control (high vs. low). The authors reported strong and consistent support for the proposed causal and negative effect of job demands on physical activity after work. The results regarding the consequences of job control were less clear as they did not show a direct or a buffering effect. The lack of representativeness of the sample, which consisted of a group of young adults (i.e., college students), was identified as a main study limitation of the study. This highlights the need to further explore the active job (high demands and high control) hypothesis in relation to physical activity and specifically in a representative sample of the population. The present study addresses these concerns in an effort to better understand how psychosocial working conditions influence behavioral change in relation to physical activity in a crisis context characterized by increased psychological demands [39] and physical inactivity [14,15]. From the perspective that workplaces can play a substantial role in the health and behaviors of individuals, this study focuses on workers who were physically inactive at the beginning of the pandemic but who became so over time. Increasing physical activity should be a cornerstone in the prevention and management of mental and physical health problems [40,41]. By understanding the role of job design in this association, this study could provide valuable insights into how organizations can help workers towards a healthy lifestyle. 

## 2. Materials and Methods

### 2.1. Sample and Design

Self-reported questionnaires were collected from a population-based web panel in the province of Quebec (Canada), including 60,000 adults (Quebec’s 2020 population of those aged 18–64 comprises 5.3 million people). A total of 6000 people was invited randomly, of whom 1450 replied that they had worked over the past seven days and agreed to participate. From these 1450 workers, a final sample of 440 physically-inactive participants was selected based on their activity level at the first assessment (T1). From a public health perspective, participants who are inactive represent the ones who are most at risk health-wise, which is why we chose to investigate the factors explaining why they became active.

A three-wave longitudinal design was used, with the length of the follow-up between the measurement times selected to compare data taken during the first lockdown in April 2020 (T1), the easing of lockdown measures during the summer in June 2020 (T2), and the second lockdown in November 2020 (T3). At T1, all nonessential businesses, including gyms, were closed in Quebec. People were permitted to play outdoor sports with people in their household. At T2, gyms and other sports facilities were reopened. As it was the summer season at that time, many outdoor sports activities, including team sports, had no limit as such to the number of participants. Finally, at T3, during the second lockdown, organized sports and leisure activities were no longer permitted, and the government again required the closure of gyms and fitness centers. Practicing outdoor physical activity in a group with a maximum of eight people was allowed as long as a distance of 2 m between people was respected.

This study was approved by the University’s Ethics Committee for Health Sciences Research, and written informed consent was obtained from each participant.

### 2.2. Measures

#### 2.2.1. Job Demands and Job Control

Job demands were assessed with five items from the short French version of the Job Content Questionnaire (JCQ) [42] and one item from the full version of the JCQ, “My tasks are often interrupted before they can be completed” [43]. Job control was evaluated with five items adapted from the JCQ [43]. A sample item is: “I have a lot of say about what happens on my job.” The response scale for both measures ranged from 1 (strongly disagree) to 4 (strongly agree). The psychometric properties of the French version of the JCQ have been previously demonstrated [44,45,46]. The median was used to dichotomize exposure to high and low job demands and the control. In this study, the reference group was the high-strain group (high demands and low control) as this represents the most deleterious psychosocial working conditions [27,28], with the highest probability of undesirable health outcomes.

#### 2.2.2. Physical Activity

The frequency and duration of physical activity were measured according to the 2014–2015 Quebec Population Health Survey. Participants were asked about activities in their free time, which can take the form of sports, fitness, outdoor activities, or simply going for a walk. Active transportation such as walking, cycling, or other means of commuting was not included (“In the past 7 days, approximately how many days did you engage in physical activities in your free time? On a typical day in the past 7 days, on average, how much time did you spend on such activities?”). The physical activity index used in the survey was constructed in line with the World Health Organization’s physical activity recommendations [47], as well as with the scientific literature on the subject [48]. In this study, participants who reported at least 150 min of physical activity per week were considered active, whereas the others were considered inactive. Physical activity was then categorized as a binary variable (0 = inactive, 1 = active).

### 2.3. Analysis

Descriptive statistics were computed for physical activity level, job demands, and job control. Inactive participants were classified into two statuses to reflect the temporal evolution of their physical activity level, based on whether they became active during T2 or T3 assessment (status = 1) or remained inactive (status = 0). Two logistic regressions were used to estimate the probability of becoming active (status = 1) based on various combinations (low vs. high) of job demands and job control at T1. Unadjusted and covariate-adjusted odds ratios were computed for each psychosocial working condition predictor. The covariates included in the analyses were sex, age, marital status (being in a relationship or married), teleworking (spending at least 80% of hours working from home during the week preceding the survey), and the presence of children at home full time (yes or no). All statistical analyses included post-stratification weights to correct for differential nonresponse (based on age, sex, rural or urban areas, education, and language) and were conducted using SAS statistical software (version 9.4; SAS Institute Inc., Cary, NC, USA) and a standard two-tailed alpha level of 5%.

## 3. Results

A total of 117 participants (out of 440, weighted % = 25.8) of the sample became physically active during the study. The proportion of participants who became physically active at Time 2 was 20.5% (*n* = 92) and 5.2% (*n* = 25) at Time 3. For those who stayed inactive throughout the three waves of the study (*n* = 323), 7% reported having an active job, 14.4% a low strain job, 55.8% a passive job, and 22.9% a high-strain job. These proportions differed from slightly the participants who became active during the study (*n* = 117); 14.1% of them reported having an active job, 17.2% a low-strain job, 49.4% a passive job, and 19.3% a high-strain job. None of the covariates were significantly associated with physical activity status, except for the covariate of having children at home full-time. Surprisingly, telework was not associated with physical activity status. Participants who reported the presence of children at home full-time were more likely to become active during the study. Descriptive statistics for predictors and covariates are displayed according to status in Table 1.

Unadjusted and adjusted odds ratios for becoming physically active according to the JDC model and covariates are displayed in Table 2. The adjusted model was found to be significant (LR test (df = 8) = 25.48, *p* < 0.001) and provided a moderate classification rate (71.6%, sensitivity = 5.2%, specificity = 96.2%). The active job (high demands and high control) had a higher unadjusted odds ratio for becoming physically active OR = 2.40 (95% CI 1.08 to 5.33).

After controlling for covariates (sex, age, marital status, teleworking, and children at home), similar results were found. An active job, compared to a high-strain job, increased the probability of becoming physically active at follow-up the most (OR = 2.57 (95% CI 1.13 to 5.87) (see Table 2)).

## 4. Discussion

In the context of a pandemic, in which the physical work and leisure environments may not be conducive to promoting a physically active lifestyle, it appears important to offer a comprehensive understanding of the psychosocial working conditions that may encourage workers to increase their physical activity level. Therefore, we examined whether exposure to an active job (high demands and high control) predicted the probability of becoming physically active during the COVID-19 outbreak. As expected, the results revealed that simultaneously having high job demands and high job control (an “active job”) increased the probability of becoming physically active compared to having high demands and low control (a “high-strain job”). These results suggest that job control may be an important psychosocial working condition of becoming active for physically inactive workers. This study offers valuable theoretical and practical implications, which we describe below.

### 4.1. Theoretical Implications

This study contributes to our knowledge in the following ways: (1) highlighting the contribution of psychosocial working conditions to individuals’ lifestyle habits and (2) demonstrating that this contribution to the workplace can still be significant in pandemic times, when opportunities for engaging in regular physical activities are limited. Indeed, our results showed that telework was not associated with physical activity status.

The present results also contribute more specifically to the JDC model. Although previous JDC-based studies have extensively documented high-strain jobs compared to active jobs, the present study provides evidence that the effect of job control goes beyond the context and boundaries of work, generating personal benefits, i.e., promoting the adoption of healthy behavior. Importantly, although most studies support the idea that certain adverse psychosocial working conditions negatively influence physical activity, we found that a subset of workers may take advantage of an active job to become physically active. As the pandemic has disrupted the lifestyles of many people, it is relevant to look at those who have seen an opportunity and have made positive changes to their habits.

Notably, our study contributes to supporting one of the propositions (proposition five; i.e., that job control buffers the negative effect of job demands on physical activity) of the pamDC model of Häusser and Mojzisch [37]. Although not directly addressed in the present study, our results suggest that job control mitigates the negative effects of high demands on physical activity. Indeed, the combination of high control and high demands is the most beneficial for physical activity. Other studies also revealed a positive association between job control and participants’ leisure-time physical activity levels, meaning that workers with more job control were more active during their leisure time [32,49,50]. As highlighted in the experimental study by Abdel Hadi et al. [33], this relationship might be explained by an indirect effect of job control on physical activity through self-determination. According to self-determination theory [51], this is because job control helps to nurture employees’ psychological needs [52]. As other mechanisms may be involved in this relationship, further studies are needed to improve this understanding. For instance, Abdel Hadi et al. [53] found that daily leisure crafting was an effective behavior that can help to decrease emotional exhaustion by recreating meaningful leisure experiences. As physical activities are known to help workers’ recovery experiences during their personal time [54], it seems important to further investigate the mechanisms that could explain why and how some employees become more or less physically active during leisure time.

Our results are partially in line with those of the meta-analysis by Fransson et al. [31]. Their results showed that the risk of physical inactivity was 26% higher in high-strain jobs (high demands and low control) than in low-strain jobs (low demands and high control). It has been suggested that high-strain jobs may cause fatigue, thus increasing the likelihood of physical inactivity. In an analysis restricted to physically active participants, the odds of becoming physically inactive during the follow-up were 21% for those in high-strain jobs at baseline. In other words, having a high-strain job increases the risk of becoming inactive over time. Our results are partly in line with these findings, as we noted that in both studies (ours and the one by Fransson et al., 2012), job control appears to be the common denominator for promoting physical activity. In the study of Fransson et al. [31], a lack of job control (combined with high demands) negatively affected physical activity, whereas in our study, high job control (also combined with high demands) positively affected physical activity. This suggests that job control mitigates the negative effects of high job demands and fosters positive health outcomes. Indeed, the winning combination of becoming physically active in our sample was high demands and high control.

The present findings enrich those of previous research on the association between psychosocial risk and physical activity, which has been characterized by mixed results. Possible reasons for this inconsistency include the disparity in how physical activity is measured and classified. Previous research has also been marked by difficulties in generalizing across geographic regions, gender, and occupation types, as these variables appear to influence the variables of interest. This longitudinal study, conducted during the first eight-month period of the COVID-19 pandemic, with its representative sample of the population according to age, sex, rural or urban area, education, and language, takes a step toward resolving these limitations. Notably, this is probably the first longitudinal study that uses a populational sample to address this important issue in the context of a pandemic, a context marked by a “natural” increase in psychological demands [39].

### 4.2. Practical Implications

To promote physical activity in the workforce, it seems essential to foster job control in the workplace. For organizations, this could be accomplished by defining the workers’ autonomy zones or developing processes and tools for participation in decisions that can guide managers and workers. Fostering the creativity of workers and teams, as well as allowing them to use their potential skills more often and to learn new skills, are other examples of how to increase job control. Other promising avenues for organizations would be to revisit aspects of job design. Of particular interest, some recent studies suggest integrating interventions into work itself, i.e., incorporating physical activity during task performance, by changing the way in which tasks are carried out [55,56,57]. Organizations could promote both job control and physical activity by providing opportunities for their employees to be physically active while working. Inspired by the Goldilocks fairy tale, Holtermann et al. [56] proposed the Goldilocks principle, which promotes a “just right” approach to designing productive work to be optimal with respect to physical activity. “In many jobs, physical activity is, however, either too much/high/frequent or too little/low/infrequent to give positive biomechanical and cardiometabolic stimuli” [56]. In other words, the Goldilocks principle aims to promote health and physical capacity by designing physical activity during productive work to be “just right.” To redesign work to better fit the Goldilocks principle, Holtermann et al. [56] proposed three types of modification: (1) changing how to perform the tasks, (2) changing the time pattern of work tasks, and (3) introducing new tasks. Those modifications represent different ways to increase both job control and physical activity.

Similarly, the study of Hervieux et al. [58] tested the concept of active meetings with seven work teams (*n* = 30). Participants were required to cycle on stationary bikes while conducting their one-hour meeting. Participants were also asked to hold one of their meetings in traditional mode (seated in a chair). The researchers compared the results of the two conditions (active and seated) and observed an increase in heartrate and a significant decrease in perceived stress during the active meetings. By letting managers and employees decide whether they actively lead their meetings, organizations would also foster decisional latitude and promote physical activity.

Because job control offers workers more action opportunities, employees are encouraged to take advantage of their time outside of work by engaging in physical activities that give them a sense of pleasure. According to Demerouti et al. [54], engaging in leisure activities does not guarantee recovery from work. Indeed, if the person does not gain pleasure or satisfaction from performing the activity, it will not contribute to recovery. As discussed by Sonnentag et al. [59], it is not a specific activity per se that helps workers recover, but rather it is the motivational and psychological attributes attached to the activities that further determine the activities’ recovery potential. Taking advantage of recovery opportunities is even more important, as a mutual relationship with working conditions has been demonstrated, suggesting that working conditions influence recovery opportunities, but recovery opportunities also influence the perception of work characteristics [60].

### 4.3. Limitations and Future Research Directions

This study has some limitations that should be acknowledged. First, although we adopted a recognized theoretical model to determine the choice of our variables, the analysis was based on a limited set of variables. Future studies are needed to enrich our understanding of the variables that increase physical activity. Regarding the covariable “presence of children at home full-time”, which was associated with an increased probability of becoming physically active, future studies are needed to document this association. Indeed, working parents could increase their physical activity level in reaction to their children’s need to be active. However, this result could also be explained by the fact that having younger children generally involves an increased domestic workload (cleaning, cooking, etc.), or that parents need to be physically active to protect their own mental health and to keep themselves fit to care for their children. Next, our study was limited to the predictors of becoming physically active during the first year of the pandemic. It would thus be interesting to investigate its effects on other indicators of physical and psychological health of employees during their work. Such information could certainly help to convince organizations to find levers of physical activity on which they can act to promote the health and optimal functioning of their employees.

Most studies have focused on two psychosocial working conditions (demands and control), although the study of other psychosocial conditions such as social support seems to provide interesting insights. Therefore, we suggest that future studies include more conditions to obtain a more comprehensive picture of the psychosocial work environment. In addition, most studies, including the present one, measured physical activity in only one domain: leisure time. The results of the studies may underestimate total physical activity time by excluding active minutes spent commuting, at work, or at home.

Although our study supports the hypothesis that the presence of high demands does not seem to be detrimental to physical activity, as long as employees have sufficient access to resources such as control, it would be interesting to examine whether there is a certain maximum level of demand that should not be exceeded. In future studies it would be helpful to quantify the threshold at which the demands become too high for the resources used to counterbalance their effects.

Future studies could investigate how self-determination at work (i.e., more intrinsic motivation) affects either the level of physical activity, its nature, intensity, or the association between psychosocial constraints and physical activity. The results obtained in this study are perhaps related to active jobs that may be characterized by higher intrinsic motivation.

Finally, since job control appears to be a resource that promotes employee health, it would be useful to investigate how employees and organizations can foster control over work in challenging times when many employees work remotely.

## 5. Conclusions

In brief, the findings of the present study show the importance of job control as a resource for positive action on health and should be one of the targets of intervention to promote physical activity. It is useful to identify psychosocial working conditions that might influence physical activity, given their importance to health and work capacity, particularly in the aftermath of a global pandemic that will inevitably affect the mental health of many workers. The accumulation of strain in a high-strain job could inhibit workers from engaging in physical activity. Having a highly demanding job may not negatively impact physical activity if workers have enough resources (e.g., job control) to achieve the required tasks. In our study, when demands are combined with a high level of control, they instead appear to promote becoming physically active. The longitudinal design of this study provides reliable results of high methodological quality. From a practical perspective, the results of this longitudinal study will help to guide interventions to support organizations that seek to promote physical activity among their employees.

## Figures and Tables

**Table 1 ijerph-19-02168-t001:** Descriptive statistics according to physical activity status.

	Stayed Inactive (*n* = 323)	Became Active (*n* = 117)	Tests *X*^2^, *t*-Test
JDC Model combinations Active (high demands, high control) Low-strain (low demands, high control) Passive (low demands, low control High-strain (high demands, low control)	7.0% (32) 14.4% (61) 55.8% (163) 22.9% (67)	14.1% (19) 17.2% (17) 49.4% (56) 19.3% (25)	6.46, *p* = 0.09
Sex Female Male	51.9% (177)	57.5% (60)	1.08, *p* = 0.30
Age (years)—mean (SD)	42.8 (12.1)	43.0 (10.0)	0.17, *p* = 0.87
Being in a relationship/married Yes No	66.2% (223)	60.8% (72)	1.08, *p* = 0.30
Teleworking Yes No	44.2% (148)	43.7% (54)	0.01, *p* = 0.93
Presence of children at home full-time Yes No	36.7% (109)	53.0% (56)	9.28, *p* = < 0.005

Note. *X*^2^ = Chi-Square; JDC = Job demand-control; weighted percentages/means and raw frequencies (*n*) are reported.

**Table 2 ijerph-19-02168-t002:** Unadjusted and covariate-adjusted odds ratios for predictors of becoming physically active at T2 or T3 (*n* = 440).

JDC Model Combinations and Covariates	Unadjusted OR (95% CI)	Adjusted OR (95% CI)
Active job	2.40 * (1.08, 5.33)	2.57 * (1.13, 5.86)
Low-strain job	1.41 (0.69, 2.87)	1.44 (0.68, 3.03)
Passive job	1.05 (0.60, 1.84)	1.09 (0.61, 1.96)
High-strain job (reference)	1.00	1.00
Sexe (female)	--	1.24 (0.79, 1.95)
Age (years)	--	1.01 (0.99, 1.04)
Being in relationship/married (Y/*n*)	--	0.52 * (0.32, 0.86)
Teleworking (Y/*n*)	--	0.98 (0.62, 1.54)
Presence of children at home full-time (Y/*n*)	--	2.71 * (1.65, 4.48)

Note. JDC = Job demand–control; ** p* < 0.05.

## Data Availability

The data that support the findings of this study are available from the corresponding author, (V.H.), upon reasonable request.

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
