# Peer review of "The Role of Job Control and Job Demands in Becoming Physically Active during the COVID-19 Pandemic: A Three-Wave Longitudinal Study"

_ijerph, 2022, doi:10.3390/ijerph19042168_

Round 1
Reviewer 1 Report
Telework and physical and psychological well-being are indeed hot topics amid pandemics and definitely need research attention. Therefore, I appreciated the opportunity to read your research entitled "The role of job control and job demands in becoming physically active during COVID-19 pandemic: A three-wave longitudinal study". I also appreciated the three-wave design, which amid pandemic is truly challenging, and that too via web panel.
However, I have several major concerns and recommendations about the paper;
- The authors have started with the demonstration of the decline in the population during the COVID-19 pandemic. Later, that physical activity is limited to organizations, even which this physical activeness is about access to the fitness centres and parks. Also, physical activity has been mostly associated with strain (high vs. low). Such that, there is no such relevance presented regarding population and physical activity. In other words, the focus of the study is distracting. Therefore, I would strongly recommend authors to frame the introduction in one direction with consistency.
- Followed by the first comment, the term physical activity itself is unclear. Authors need to provide a specific definition of the term according to the context which readers could follow throughout the content.
- Line 110- Since authors have talked about self-regulation and self-determination, they should consider reading this piece of paper on emotional labor through the lens of determination theory (Cossette, 2014) and the role of leisure crafting for emotional exhaustion in telework during the COVID-19 pandemic (Abdel Hadi et al., 2021). These will help them to strengthen the arguments more appropriately.
- Regarding measures, “On a typical day in the past 7 days, on average, how much time did you spend on such activities?”. Could authors mention which “such activities” were asked to rate?
To summarize, I would encourage you to work on the writing, rationality, consistency, and flow. Particularly, the writing is not at a level I would expect to see in a published journal paper. Perhaps good editing could help (also with the language e.g., line 122). Good luck!
Abdel Hadi, S., Bakker, A. B., & Häusser, J. A. (2021). The role of leisure crafting for emotional exhaustion in telework during the COVID-19 pandemic. Anxiety, Stress and Coping, 0(0), 1–15. https://doi.org/10.1080/10615806.2021.1903447
Cossette, M. (2014). Emotional labor through the lens of self-determination theory. The Oxford Handbook of Work Engagement, Motivation, and Self-Determination Theory, Oxford University Press, New York, NY, 259–275.
Author Response
|
Reviewer #1’s comments |
Our response |
|
1. The authors have started with the demonstration of the decline in the population during the COVID-19 pandemic. Later, that physical activity is limited to organizations, even which this physical activeness is about access to the fitness centers and parks. Also, physical activity has been mostly associated with strain (high vs. low). Such that, there is no such relevance presented regarding population and physical activity. In other words, the focus of the study is distracting. Therefore, I would strongly recommend authors to frame the introduction in one direction with consistency. |
We have rewritten the introduction considerably to focus on a single direction, which is based on gaps in the scientific literature on the relationship between psychosocial working conditions and physical activity. We removed most of the passages that referred to the problem of the inaccessibility of the physical environment for participating in physical activity. |
|
2. Followed by the first comment, the term physical activity itself is unclear. Authors need to provide a specific definition of the term according to the context which readers could follow throughout the content. |
To clarify the term physical activity, we gave the definition we used in the Introduction. |
|
3. Line 110- Since authors have talked about self-regulation and self-determination, they should consider reading this piece of paper on emotional labor through the lens of determination theory (Cossette, 2014) and the role of leisure crafting for emotional exhaustion in telework during the COVID-19 pandemic (Abdel Hadi et al., 2021). These will help them to strengthen the arguments more appropriately. |
Thank you for suggesting these interesting papers. We indeed add some insights of the study by Abdel Hadi, Bakker & Häusser (2021) in the Discussion section (Theoretical Implications), as we suggest future studies should investigate mechanisms such as leisure crafting to better understand the relationship. We thank the reviewer for this interesting suggestion but felt that emotional labor as a topic was somewhat outside the scope of this paper, but we however integrated insights from the other reference suggested.
|
|
4. Regarding measures, “On a typical day in the past 7 days, on average, how much time did you spend on such activities?”. Could authors mention which “such activities” were asked to rate? |
This has been clarified on p. 4 (see lines 183-286). |
|
5. To summarize, I would encourage you to work on the writing, rationality, consistency, and flow. Particularly, the writing is not at a level I would expect to see in a published journal paper. Perhaps good editing could help (also with the language e.g., line 122). |
As suggested, we have reworked the introduction and discussion sections to improve the flow and clarify of the contribution. In addition, we also have added a new section “Limitations and Future Research Direction). Note that this revised version of the paper has been carefully revised by a professional language editing service. The certificate of the revision undergone by MDPI is attached. |

Reviewer 2 Report
The researchers analyzed psychosocial variables in relation to working conditions during the pandemic, intending to establish a link with psychophysiological aspects. This psychophysiological expression would be most appropriate for this research, since the implications have effects on the respondents' physical health. It's a good contribution, just a little obvious, but rigorously tested. The statistical analysis has been performed extremely well; there is indeed a high level of methodological rigor that ensures the integrity of the studied data. In the background, it was mentioned that psychosocial elements, as they are described, contribute to enhanced health but are mostly unexplored. I have doubts regarding whether they were really unexplored. Has any prior scientometric research been conducted to ensure this? A systematic review? Due to the manner in which the text was written, it is essential to improve on these points. The conclusion brings up the question of whether this study satisfies the initial research GAPs that remain unexplored.Additionally, there are no allusions to future GAPs, which could contribute to future research. We know that working conditions have an effect on how people behave when it comes to physical activity. This was amplified throughout the pandemic. This moment was characterized by a high demand for psychosocial difficulties and physical inactivity, both of which have an effect on psychophysiological health. However, how could we maintain control over the work? This could be a future GAP. Despite the fact that I have expressed doubts, this has no implications on the quality of the research, which I recommend for publishing.
Author Response
|
Reviewer #2’s comments |
Our response |
|
1. The researchers analyzed psychosocial variables in relation to working conditions during the pandemic, intending to establish a link with psychophysiological aspects. This psychophysiological expression would be most appropriate for this research, since the implications have effects on the respondents' physical health. |
This expression is appropriate and has been added to “Limitations and Future Research Direction” section. |
|
2. In the background, it was mentioned that psychosocial elements, as they are described, contribute to enhanced health but are mostly unexplored. I have doubts regarding whether they were really unexplored. Has any prior scientometric research been conducted to ensure this? A systematic review? Due to the manner in which the text was written, it is essential to improve on these points. |
As suggested, we have reworked the introduction and discussion sections to improve the flow and clarify of the contribution What remained unexplored was the psychosocial working conditions contributing to becoming physically active (versus adverse psychosocial working conditions associated with physical inactivity) during the pandemic where work conditions were disrupted. This has been clarified on pp.1-2 (see lines 36-40). |
|
3. The conclusion brings up the question of whether this study satisfies the initial research GAPs that remain unexplored.Additionally, there are no allusions to future GAPs, which could contribute to future research. We know that working conditions have an effect on how people behave when it comes to physical activity. This was amplified throughout the pandemic. This moment was characterized by a high demand for psychosocial difficulties and physical inactivity, both of which have an effect on psychophysiological health. However, how could we maintain control over the work? This could be a future GAP. Despite the fact that I have expressed doubts, this has no implications on the quality of the research, which I recommend for publishing. |
As suggested, we have added a new section “Limitations and Future Research Direction” in which we also suggest investigating how employees and organization can foster control over work in those challenging times where many employees work remotely. |
Reviewer 3 Report
The evaluated study aimed to examine whether exposure to different combinations of job demands and job control predicts the probability of becoming physically active during the first months of the COVID-19 outbreak, which is of special importance for improving a resistance against virus SARS-CoV-2. The interesting methodological approach was applied. The interesting and easily understandable result was that participants who reported the presence of children at home full time were more likely to become active during the study.
Remarks:
1) It would be worth supplying the Introduction with some additional lines about positive effects of the regular physical activity on cardiovascular system, glucose and lipids serum levels, and immune system. 2) Contrary to this, the subsection 1.1. seems to contain too many deliberations; a large part of them should be included rather in the Discussion section. 3) According to the methodology used, in relation to the physical activity changes, the authors analyzed only the levels of job demands and job control, however it might be interesting to take into consideration also at least a general characteristics of professions taken by the participants. Adding such an information would give a deeper insight into the searched correlations.
Author Response
|
Reviewer #3’s comments |
Our response |
|
1. It would be worth supplying the Introduction with some additional lines about positive effects of the regular physical activity on the cardiovascular system, glucose and lipid serum levels, and immune system. |
The introduction has been improved with additional information about the positive effects of regular physical activity on cardiovascular system, mental health and physical function (see lines 46-48). |
|
2. Contrary to this, the subsection 1.1. seems to contain too many deliberations; a large part of them should be included rather in the Discussion section. |
A portion of the sub-section has been removed and another portion has been moved to the Discussion section. |
|
3. According to the methodology used, in relation to the physical activity changes, the authors analyzed only the level of job demands and job control, however it might be interesting to take into consideration also at least a general characteristic of professions taken by the participants. Adding such an information would give a deeper insight into the searched correlations. |
Supplementary analyses, conducted while preparing this revision, indicate that profession’s type did not differ (Chi-Square = 5.8292 p = .32) from those who became active and those who stayed inactive. Moreover, we re-estimated our main analyses while adding profession’s type as a control variable and it had no impact on our results.
Adjusted odd ratios for the JDC model combinations with profession’s type added as a control variable = Active job 2.32 Low strain job 1.29 Passive job 1.11 High strain job (ref). 1.00
Since the data were collected during the pandemic, and during two lockdowns (T1 and T3), it is plausible that teleworking was a variable that took precedence over occupation type, as the latter had no effect on our results. During the pandemic and within our sample, it would therefore be possible to believe that it is more the fact of having a job that can be done by teleworking, rather than the profession per se that influences the results.
|
Reviewer 4 Report
- The article seems interesting and the research was conducted on a representative research sample.
- The research topic is not groundbreaking and difficult to understand, due to the dependence of many variables that were not necessarily included in the study. There is no clear justification as to why this research topic was undertaken in the context of the research gap. A complete survey questionnaire is also missing.
- Unfortunately, the article lacks research assumptions confirmed by the analysis of the literature. The study takes into account the effect of physical activity during a pandemic, but other variables not included in the analyzes may also affect the increase in activity of study participants. This fact should be taken into account in the literature review and rigorous research assumptions should be adopted.
- The strain jobs was analyzed in the research - it was not defined whether it was related to the period before the locdown, or in general. This change may be significant and is not accounted for in the study or in the discussion of results / assumptions.
- The study indicated that the respondents are prone to be more active when they have children. Isn't it the assumption that children need activity and that parents adjust to it?
- The research results were presented in a sufficient manner and nicely in connection with the research results obtained so far and described in the literature.
- The research results are interesting especially in the practical aspect.
Author Response
|
Reviewer #4’s comments |
Our response |
|
1. The research topic is not groundbreaking and difficult to understand, due to the dependence of many variables that were not necessarily included in the study. There is no clear justification as to why this research topic was undertaken in the context of the research gap. A complete survey questionnaire is also missing. |
We clarified why this research topic was undertaken in the light of research gaps in the introduction (see lines 32-39).
We are not sure what you mean by “A complete survey questionnaire is also missing.” If you meant by that the questionnaire completed the participants, we are sending it in a supplement material. |
|
2. Unfortunately, the article lacks research assumptions confirmed by the analysis of the literature. The study takes into account the effect of physical activity during a pandemic, but other variables not included in the analyzes may also affect the increase in activity of study participants. This fact should be taken into account in the literature review and rigorous research assumptions should be adopted. |
We agree that other variables not included in the analyses may also affect the increase in physical activity of the participants. We acknowledged that statement in the new Limitations and Future Research Direction section. |
|
3. The strain jobs was analyzed in the research - it was not defined whether it was related to the period before the lockdown, or in general. This change may be significant and is not accounted for in the study or in the discussion of results / assumptions. |
In the section 2.1 we had specified that the first assessment of our study was during the first lockdown in April 2020, which is approximately one month after the beginning of the pandemic. |
|
4. The study indicated that the respondents are prone to be more active when they have children. Isn’t it the assumption that children need activity and that parents adjust to it? |
In fact, this can be one of the reasons that might explain this result. Another reason that might explain this result is that as participants with children at home might increase their domestic physical activities such as picking up toys, doing more laundry, etc. However, as we did not assess this dimension of physical activity, those remained only hypothesis. We acknowledge this in the new “Limitation and Future Research Direction” section. |

Round 2
Reviewer 1 Report
Thanks for the authors' efforts to revise the manuscript and for being responsive to my comments. I have some minor comments below.
- There is considerable evidence suggesting that psychosocial working conditions, such as low job control and high job demands, are important predictors of job strain (see; author 2000; etc). It is better to cite or refer to such evidence.
- The first paragraph in the introduction still sounds like an argument, not a rational phenomenon. I suggest the authors frame it as a storyline that readers/general audience can relate to and take them to the gap which authors intend to fill. It would be better if they start with the importance of healthy lifestyle/physical activities/working conditions.
Thanks again for your efforts to revise the manuscript and I hope the comments are helpful to improve the manuscript further.
Author Response
Thank you for you relevant suggestions.
As suggested, we add a phrase at the beginning of the Introduction to emphasize the importance of having healthy and positive working conditions to foster healthy behaviors among employees.
Two references (including one systematic review) were added in the second sentence to support the statement.